# The Most Frequently Cited Topics in Urban Planning Scholarship

**Thomas W. Sanchez**

Urban Affairs and Planning, Virginia Tech, Arlington, VA 22203, USA; tom.sanchez@vt.edu; Tel.: +1-540-231-5485

**Abstract:** Analyses of faculty citation activity usually focus on counts as a function of author characteristics, such as rank, gender, previous citation levels, and other factors influencing productivity and career path. Citation analyses of publications consider aspects, such as the number of authors, author reputation, author order, length of the title, methodology, and impact factors of the publication. While publication topics or discipline is considered important factors, they are more difficult to analyze, and therefore, performed less frequently. This article attempts to do that for the field of urban planning. Urban planning is multi-disciplinary and includes consideration of social, economic, technological, environmental, and political systems that shape human settlement patterns. It has been suspected that some topics are more "popular" and have larger audiences, therefore, are cited more often. Using nearly 15,000 urban planning publications, this article presents an analysis of topics to assess which are cited most frequently. The classification of publications was performed using a Support Vector Machine (SVM), a machine learning (ML) approach to text classification, using citation data from Google Scholar. The citation levels for the resulting categories are analyzed and discussed.

**Keywords:** urban planning; bibliometrics; citation analysis

## 1. Introduction

Urban planning is an interdisciplinary field, focused on understanding human settlement patterns [1]. Planning is a specialized and relatively small discipline that draws on design (e.g., architecture), policy (e.g., public administration), and social science (e.g., geography). Like other social sciences, urban planning scholars disseminate their research in a wide array of academic journals. For just over 1100 urban planning faculty in the USA and Canada, there are over 300 journals where their publications appear. It is difficult to say whether this is a significantly large number of outlets without similar metrics from other disciplines. However, the objective of this article is to examine which research topics are of interest to current urban planning faculty in North America, and not a comparison to other disciplines. The purpose is to identify prominent topics so that planning academics can reflect on whether these are the issues that should be receiving our attention. The analysis concentrates on faculty in urban planning programs and their publication activities, and not what is being published about urban planning topics in general. This provides the opportunity for planning to consider its scholarly priorities as we look to the future of urban planning and urban science research.

With urban planning interests representing several disciplines, we suspect that there is a shifting hierarchy of topics over time. These include, but are not limited to, changing perspectives on infrastructure policy, architecture, social conditions, environmental conditions, economic activity, and governance—all of these representing significant aspects of urban systems. Not all of these are given equal attention over time, as particular concerns or interests are more visible at certain times compared to others. From the public's perspective, these interests are being driven by changing policies or politics, and by the flow of information from sources, such as the news media. From an academic and institutional standpoint, themes of research are certainly influenced by the availability of funding and

the priorities of funding organizations [2,3]. While there are no comprehensive data sources about funded research activities related to urban planning, it would be interesting to know how or whether these topics differ from those of scholarly output. This remains a topic for future research.

Within urban planning, there is also the question of how academic research represents the needs of the practicing professionals [4,5]. Planning professionals are confronted with new and different types of urban questions, as well as seeking solutions to on-going questions. Loh [6] argues that "planning is very much an action-oriented field" and therefore, research "not closely focused on what planners actually do is of limited relevance to the profession" (p. 25). Urban planning scholars have long contributed knowledge to the planning profession through their teaching, research, and service activities, while at the same time meeting traditional academic expectations through scholarship [7]. Planning students seek professionally-oriented training, and academic planning programs are expected to connect directly with local communities through their service and outreach activities [8,9]. However, the question remains whether planning academics are more responsive to topics when funding is available, when research is needed by the profession, or when the topic is of academic interest. Some have argued that placing scholarship as a priority over teaching and service in planning programs is a result of institutional pressures for faculty having doctorates rather than professional certification. As Krumholz noted, the emphasis on "a productive history of publication in refereed journals," contributes to a divide between planning research and practice [10].

*Citation Analyses*

Citation analysis has an extensive literature documenting citation practices across disciplines, as well as meticulous discussion of measurement and evaluation [11–14]. Citation analysis is a way to evaluate scholarship to gauge prominence and productivity, but often excluding other dimensions, such as visibility and impact [15]. Citation analysis is also benefitting sophisticated text analytics techniques that move beyond extracting and counting citation to determining context and intent for citations [14]. The following provides a brief discussion and background on citation data, citation analysis, and urban planning scholarship.

This analysis uses citation metadata and counts from Google Scholar (GS). Many researchers have examined GS for citation analyses by making comparisons to the other prominent sources of citation data (such as Scopus and Web of Science (WoS)). One concern that has arisen is about GS's coverage of scholarly publications relative to that of Scopus or WoS [16–19]. However, these issues are likely discipline-specific, with many examples highlighting GS's reliability for topics ranging from oncology and condensed matter physics [20], business and economics [21], health and medical research [22]. Meta-analyses are useful for illustrating patterns in bibliometric performance by examining different data sources and analytical methods [23,24]. Many of the analyses comparing GS with Scopus and WoS concentrate on the difference in the number of citations rather than the accuracy of these data at the for individual authors. However, to assess the accuracy of citation counts for an author, a verified list of publications, such as from a current CV (Curriculum Vitae) would have to be compared to those listed in citation databases. This is not presently feasible because there are no centralized sources of confirmed CV data or publication records that can be matched to those in citation databases.

GS includes "non-traditional" publications unlike Scopus and WoS [25,26]. The presence of non-peer-reviewed publications in GS is a relevant question, when considering scholarly citation counts. The inclusion of gray literature, for instance, has been argued to have a greater reach and impact compared to pay-wall-protected publication and citation data from major publishers. Professionally oriented disciplines like urban planning produce gray literature that is research-based and reflects the scholarly process of particular value to allied professions, and therefore, reflects academic rigor and impact [27–29]. Finally, Pauly and Stergiou [30] stated that "free access to these data provided by Google Scholar offers an avenue for more transparency in tenure reviews, funding and other science policy issues, as it allows citation counts, and analyses based thereon, to be performed and duplicated

by anyone" (p. 2). This is a strong case for GS as a source of citation analysis and the fact that it includes gray literature is particularly pertinent to planning academics.

As mentioned earlier, urban planning draws upon a diverse range of disciplines and expertise. These form urban planning subfields that are reflected in the types of publication topics, as well as the breadth of journals where these publications appear. Of the few journal articles about urban planning citation activity beginning with Stiftel, Rukmana, and Alam [31] (followed several years later by Sanchez [32], Pojani et al. [33] and Stevens et al. [34]), only Stevens et al. have explicitly tried to measure topical differences within urban planning citation patterns. In their analysis of factors affecting urban planning citations, they examined whether the publication topics were related to thirteen selected topics. They found that compared to "transportation", nearly all of the other 12 were cited less frequently. Compared to the current analysis, Stevens et al. had a sample of 580 compared to nearly 15,000 in this analysis. In addition, Stevens et al. used manual, single-label topic classification versus multi-label in this analysis. The sample size issue could introduce bias by underrepresenting certain planning topics.

The bibliometric literature has recognized the differential rates of citation by topics, following the assumption that certain sub-fields are more popular, have more publications, and therefore, greater chances of citation [35]. One methodological issue is how to classify or categorize publications so that citation rates can be accurately compared. This includes an approach to normalize citation counts based on some of these factors [36]. These methods include topic analysis [37], author-provided keywords [38], thematic analysis [39], or based on categorization defined at submission by authors or journal editors. Topics and keywords restricted by journals can have a limiting effect on topic or theme classification through unintentional exclusion. An alternative would be to analyze abstracts for these purposes, but it greatly increases the data collection and analysis task for the uncertain benefit to the classification process. In addition, titles are consistently available across publication types (i.e., journal articles, books, or reports) where abstracts and keywords are not. Therefore, titles provide a rich data source for the current analysis.

The title of a publication is considered to be important for not only indicating content, but also for attracting attention [40,41]. While the author provided keywords are frequently used for discovery, titles themselves can distill several dimensions of a publication, including subject, method, geographic context, and results [42]. Because distinct keywords may not convey an overall theme, such as what can be represented in the sentence form of a title. This has to do with the linguistic structure of a title not intended by keywords [43]. However, Levy and Ellis [44] suggest that the author provided keywords may be buzz-word laden, and perhaps unreliable in the long-term. In addition, besides the information retrieval aspects of titles, bibliometricians have examined how title characteristics correlate with citation rates [45–47]. Such analyses include title length, punctuation, structure, use of acronyms, and descriptiveness. However, in cases where title attributes are correlated with higher citation rates, it has not been suggested that this somehow indicates a higher level of publication quality.

## 2. Methodology

This analysis first collects publication records from current planning academics with Google Scholar Citation Profiles. At the time of this analysis, 598 out of 1109 planning faculty in USA and Canada had GS profiles (54%), and their publications accounted for over 75% of total citations (as of June 2019) by planning faculty. The next step identified topics based on word and term frequencies from publication titles. This was conducted using thematic analysis within NVivo (QSR International, Melbourne, Australia). The process involves text analysis to build nodes of related words and terms to identify themes [48] for background on the thematic analysis). During the first step, NVivo automatically identified 102 nodes, which were then manually reviewed and recombined to obtain the resulting 30 themes. This is challenging in some cases for terms like "analysis" which can appear on its own as a topic about methodologies, but often attached to a particular subject of analysis, such as "transportation analysis", "economic analysis", "environmental analysis", etc. Because the

classification of publication titles uses multiple labels, these associations are expected to emerge from SVM labelling. The result of this analysis was the 30 key topics shown in Table 1. All of the publication titles from faculty GS profiles were then coded using the topics (labels) with a Support Vector Machine (SVM).

**Table 1.** Nodes from thematic analysis (in alphabetical order).

| Topics | | |
| --- | --- | --- |
| Analysis | Governance | Policy |
| Change | Hazards | Public |
| Cities | Health | Regional |
| Community | Housing | Social |
| Design | Impacts | Spatial |
| Development | Land Use | Sustainable |
| Economic | Local | Systems |
| Engagement | Management | Technology |
| Environmental | Neighborhood | Transportation |
| Global | Planning | Urban |

Themes can be identified through the classification of full text, abstracts, or titles. Full text, abstracts, and titles obviously contain different levels of detail given the amounts of data (total words) associated with each. In this case, the purpose was to identify fairly high-level groupings of topics, so titles were used for this because they are intended to be a distillation of publication content as previously discussed. This parallels the work of Sanchez and Afzalan [49] who used self-reported areas of research interest from urban planning faculty (see Table 2 for the list of top 30 research interests). Comparing Table 1; Table 2 suggests that in the case of urban planning publications, titles are useful sources of this information, with significant similarity in topic identifiers between stated research interests and publication titles. There were 21 of 30 faculty research interests that matched the topics from the thematic analysis, while others were very closely related, such as "studies" and "analysis"; "spatial" and "GIS"; "participation" and "engagement"; and "growth" and "change". The comparison of these two lists helped to confirm the appropriateness of the labels derived from the thematic analysis.

**Table 2.** Top 30 urban planning faculty research interests from Sanchez and Afzalan (2017) (in alphabetical order).

| Labels | | |
| --- | --- | --- |
| Analysis | Growth | Planning |
| Change | Health | Policy |
| City | History | Public |
| Community | Housing | Regional |
| Design | International | Social |
| Development | Land-use | Studies |
| Economic | Management | Sustainable |
| Environmental | Methods | Theory |
| Finance | Neighborhood | Transportation |
| GIS | Participation | Urban |

*Support Vector Machine*

Machine learning (ML), natural language processing (NLP), and text classification are being applied to a wide variety of unstructured data. These include sentiment analysis of customer comments, recommendation systems, and web page classification. Text analysis is used to find word usage patterns that help to identify themes or clusters, especially from very large data sets. Data sets can be manually labelled, which can become infeasible when the volume of words (data) or records reaches the thousands, millions, or billions. In addition, automating this process with machine learning can

also ensure that labels are assigned in a consistent manner [50]. This analysis used the support vector machine (SVM) approach to classify all publication titles using the labels shown in Table 1.

A SVM is a machine learning algorithm used for object classification. The process maps observations in high-dimensional space which are then partitioned by "support vectors" that optimize how associated points are distinguished from others. The process typically involves training the algorithm by using previously classified records so the algorithm can "learn" the differences among classes (i.e., groups) [51]. Training usually occurs during manual classification or a "supervised" stage, and once trained, the algorithm can classify other observations automatically, during an "unsupervised" stage. The algorithm can also be improved as items are validated [52,53]. This analysis used the SVM platform, MonkeyLearn, to classify the text (titles) using the 30 labels shown in Table 1. This involved supervised training of the model with at least 50 observations for each label. The training process involved manual inspection of labelled features to ensure accuracy and increase the confidence of the model. The SVM model creates an algorithm that is then applied consistently across the remaining (unlabeled) features. A sample of 100 labelled titles was randomly selected and manually tested for accuracy. Of the 100 titles, 92% of the labels were assessed as being correct. Overall, there were 272 labels applied to this sample, averaging 2.7 labels per title.

After all publication titles were classified, citation frequencies for the labelled themes were analyzed. The groups of publications were compared in terms of: (1) Total publications; (2) total citations; and (3) the average citations per publication per year. The total citations for each topic were normalized by the total number of publications and the age of the publication in years. This is important because topics with more publications will likely have the most citations, and older publications had more time to accumulate citations.

The process classified publication titles based on multiple labels (e.g., "transportation" and "housing"), instead of single labels for the more effective topic specification. This accounts for the nature of planning publication topics and title structure that often reflect the subject, method, and results of a publication. This differs from the single-label approach used by Stevens et al. (2019) as mentioned earlier. For instance, a publication about transportation systems that consider land-use activities is not just about transportation or just about land-use. In a case like this, a multi-label, SVM assigns the two labels, "transportation" and "land-use". Figure 1 shows the distribution of 30 topic labels. Each row shows the proportion of co-labeled topics with the size of the dot showing the relative frequency for each dyad. The first row shows that "analysis" was co-labeled with most of the other themes with similar frequencies, with the exceptions of "governance" and "engagement." "Economic" and "environment" had similar patterns and were frequently co-labeled with most of the other topics. Each row is interpreted as the overall number of times the corresponding row label has appeared with the column labels in any combination. Other noteworthy cases are "housing" and "land use", "planning" and "engagement", "policy" and "governance", "policy" and "local", and "social" and "health". These dyads make sense and represent population planning research topics.

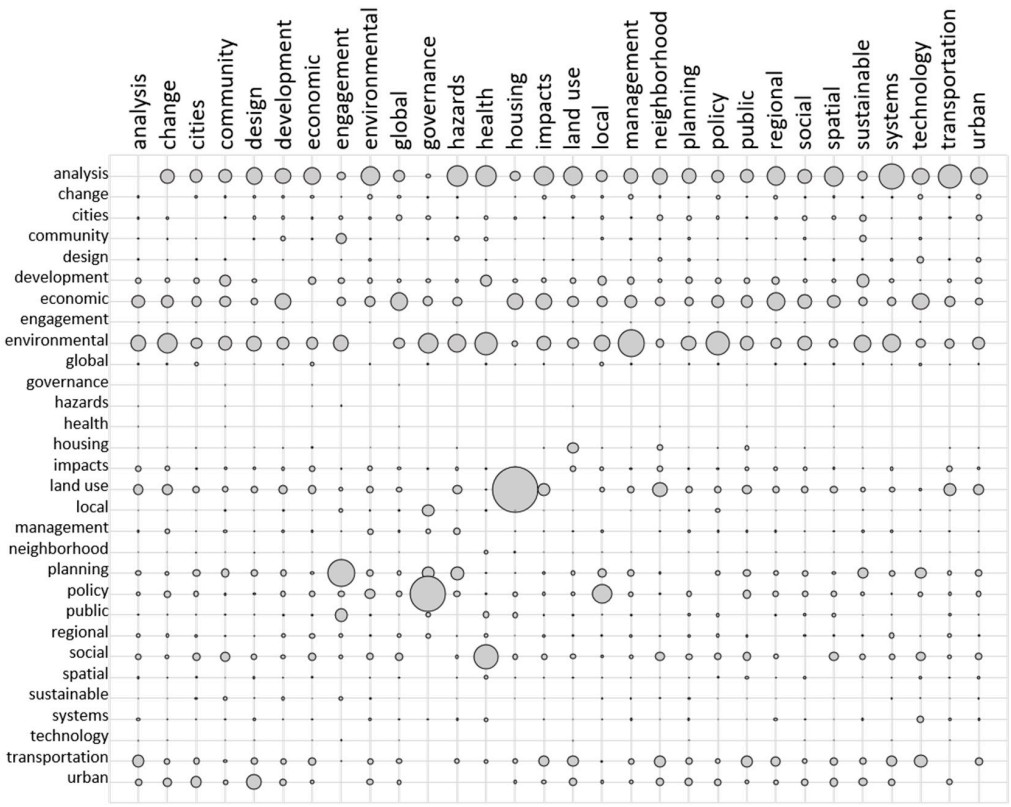

**Figure 1.** Co-labelling distribution across publication themes.

## 3. Results

Using 14,757 publications titles retrieved from 598 Google Scholar Citation profiles, the method generated 3674 topics or themes. While this may sound like a large number of topics, the 30 classifiers resulting from the thematic analysis means that there were over 1 billion possible combinations of these 30 topics ($2^{30}$ = 1,073,741,824). These themes resulted from the assignment of multiple labels using the SVM approach described earlier. The top 20% of publications (by citation totals) represents nearly 80% (79.8%) of total citations in the sample. Sanchez [32] found that the 80-20 rule (also referred to as Lotka's Law) applied in the case of faculty citations, where the top 20% of faculty generated 80% of all citations. Table 3 shows the 20 themes having the highest levels of citation activity.

**Table 3.** Top 20 topics by mean cites per publication per year.

| Rank | Topic | Number | Total Cites | Mean Cites | Mean Cites/Year |
|------|-------|--------|-------------|------------|-----------------|
| 1 | Development Economic Global Regional | 3 | 4874 | 1624.7 | 76.8 |
| 2 | Change Management Transportation | 5 | 5539 | 1107.8 | 75.5 |
| 3 | Policy Spatial Urban | 4 | 3779 | 944.8 | 67.6 |
| 4 | Environmental Social Systems | 4 | 2571 | 642.8 | 58.6 |
| 5 | Analysis Urban Environmental Social Systems | 3 | 1189 | 396.3 | 36.2 |
| 6 | Cities Urban Environment | 9 | 1526 | 169.6 | 29.7 |
| 7 | Environmental Planning Policy Analysis | 6 | 1250 | 208.3 | 29.0 |
| 8 | Impact Analysis Urban Systems | 3 | 1064 | 354.7 | 28.7 |
| 9 | Sustainable Community Policy | 3 | 891 | 297.0 | 23.5 |
| 10 | Urban Social Change | 4 | 828 | 207.0 | 22.4 |
| 11 | Urban Environmental Systems | 4 | 580 | 145.0 | 22.1 |
| 12 | Land Use Transportation Impacts | 11 | 3414 | 310.4 | 20.6 |
| 13 | Social Sustainability | 3 | 408 | 136.0 | 19.5 |
| 14 | Global Regional Economics | 9 | 1969 | 218.8 | 19.5 |

**Table 3.** *Cont.*

| Rank | Topic | Number | Total Cites | Mean Cites | Mean Cites/Year |
|------|-------|--------|-------------|------------|-----------------|
| 15 | Urban Environmental Land Use Impacts | 3 | 589 | 196.3 | 19.3 |
| 16 | Cities Environmental Change | 4 | 659 | 164.8 | 18.4 |
| 17 | Sustainable Development Planning Analysis | 3 | 927 | 309.0 | 18.3 |
| 18 | Design Impacts Analysis | 3 | 755 | 251.7 | 17.4 |
| 19 | Housing Land Use Environmental Planning | 12 | 964 | 80.3 | 17.0 |
| 20 | Social Design | 11 | 762 | 69.3 | 16.7 |

The results show that key topics in planning publications include economics, analysis, environment, transportation, regional, social, planning, and urban. These are the core elements underlying geographic scale (urban and regional), urban systems (economic, environment, social, mobility), and methodology (planning and analysis). Of the 30 identified topics, analysis, environment, economic, and transportation were the most frequently cited with each having over 100,000 citations (see Table 4).

**Table 4.** Topic frequency (in single or multiple topics).

| Topic | Number | Total Cites | Mean Cites | Mean Cites/Year |
|-------|--------|-------------|------------|-----------------|
| Global | 795 | 51,382 | 64.6 | 5.3 |
| Urban | 1897 | 96,802 | 51.0 | 5.1 |
| Spatial | 688 | 34,816 | 50.6 | 5.0 |
| Regional | 972 | 59,964 | 61.7 | 4.7 |
| Cities | 1179 | 51,755 | 43.9 | 4.6 |
| Governance | 20 | 894 | 44.7 | 4.5 |
| Neighborhood | 280 | 11,969 | 42.8 | 4.3 |
| Systems | 697 | 28,893 | 41.5 | 4.3 |
| Environmental | 3739 | 161,408 | 43.2 | 4.1 |
| Social | 1935 | 77,942 | 40.3 | 4.0 |
| Sustainable | 474 | 17,932 | 37.8 | 4.0 |
| Change | 951 | 40,893 | 43.0 | 3.9 |
| Transportation | 2609 | 100,091 | 38.4 | 3.9 |
| Health | 20 | 498 | 24.9 | 3.6 |
| Land Use | 2465 | 91,732 | 37.2 | 3.5 |
| Analysis | 4787 | 171,487 | 35.8 | 3.5 |
| Policy | 1442 | 56,212 | 39.0 | 3.5 |
| Impacts | 1293 | 42,989 | 33.3 | 3.5 |
| Management | 823 | 35,366 | 43.0 | 3.4 |
| Planning | 1756 | 61,550 | 35.1 | 3.3 |
| Housing | 532 | 18,097 | 34.0 | 3.3 |
| Economic | 3650 | 145,779 | 39.9 | 3.3 |
| Design | 803 | 28,213 | 35.1 | 3.3 |
| Development | 1699 | 68,765 | 40.5 | 3.3 |
| Local | 402 | 14,618 | 36.4 | 3.1 |
| Public | 696 | 21,452 | 30.8 | 3.0 |
| Community | 757 | 26,347 | 34.8 | 2.9 |
| Hazards | 42 | 1776 | 42.3 | 2.9 |
| Engagement | 20 | 553 | 27.7 | 2.3 |
| Technology | 28 | 387 | 13.8 | 1.7 |

## 4. Discussion

A key finding of this analysis is that instead of particular topics having high rates of citation activity, differences may be driven by particular, highly cited publications. Comparing the multi-label results in Table 3 with the single-label results in Table 4, it appears that as classification becomes more specific, particular highly cited publications influence the identification of a topic. The top 20 topics only averaged 5.4 publications compared to nearly 1250 for the 30 themes identified. The citation rates

also differ significantly. For instance, the leading topic, "Development Economic Global Regional" averaged 76.8 citations per publication per year, and the top theme, "Global" averaged 5.3 citations per publication per year. Examples from the top 5 in Table 3 (averaging over 30 citations per year) include those shown in Table 5. Given the small number in each topic area, it is easy to see how the average number of citations are influenced by particular publications.

**Table 5.** Examples of highly cited publications.

| Theme | Publication | Cites | Cites/Year |
|-------|-------------|-------|------------|
| Development Economic Global Regional | Storper, M. (1997). *The regional world: Territorial development in a global economy*. Guilford Press. | 4728 | 225.1 |
| Change Management | Feldman, M. S., & Pentland, B. T. (2003). Reconceptualizing organizational routines as a source of flexibility and change. *Administrative science quarterly*, *48*(1), 94–118. | 2988 | 199.2 |
| Policy Spatial Urban | Brenner, N. (2004). *New State Spaces*: *Urban Governance and the Rescaling of Statehood. Oxford University Press.* | 3674 | 262.4 |

## 5. Conclusions

The purpose of this analysis was to illustrate differential citation levels across topics within urban planning scholarship. The analysis presents a snapshot of scholarship, drawing from nearly 50 years of publications, authored by current urban planning faculty in the USA and Canada, in order to highlight topics receiving the most attention in a number of publication and citation activity. As demonstrated, the approach to identifying topics can impact the types of topics being compared, especially related to the level of specificity. It is possible that using more data (such as abstracts or full-text) could lead to somewhat different and more fine-grained groupings of topics because using more words may end up producing more labels or categories. Overall, the analysis showed which topics have generated the most interest from planning academics and those of planning and related fields.

It would not surprise urban planning scholars that the highest level of publication activity occurs on the topics related to analysis, environment, economy, transportation, and urban. This is because these are areas with substantial audiences and publication opportunities, both within urban planning and related fields, such as environmental studies, economics, civil engineering, and urban studies. An objective of the article was to report citation activity, but this also points to a limitation of the analysis. As mentioned earlier, citation levels are a function of several factors and not just based on publication topic or content. Title structure, author prominence, journal status, and other characteristics of visibility should be controlled for in future analyses of these data. The results reported here are not normalized by the factors mentioned above. Nonetheless, the patterns of urban planning topics and citation activity provide useful information not previously assembled at this level of specificity.

Future analyses using these types of data can take many forms. This analysis used a relatively general level of topic analysis arriving at 30 single word topics. Subsequent analyses could be more granular to be more specific about topics of scholarly activities. For instance, a topic like transportation could instead be further divided into sub-topics, such as freight, passenger, impacts, service planning, etc. However, as mentioned earlier, these topics may not be the best predictor of high citation activity, but rather other characteristics of individual publications.

**Funding:** This research received no external funding.

**Acknowledgments:** I want to thank the three reviewers for their excellent comments. Thanks also to Lucas Mun of Virginia Tech for his valuable programming assistance.

**Conflicts of Interest:** The author declares no conflict of interest.

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
