# Peer review of "The Most Frequently Cited Topics in Urban Planning Scholarship"

_urbansci, doi:10.3390/urbansci4010004_

Round 1

Reviewer 1 Report

Great paper describing the most cited topics in Google Scholar for the field of urban planning. Paper is well written, clear and concise. I did not detected any language problem.

Overall the information i well reported, tables are not so visual appealing. This paper could benefit from nice diagrams to present information.

For some reason Table 1 jumped from pg. 5 to 6, this can be fixed. Still table 1: It is odd that the systematic used in the paper did not find topics such as mobility, participation, bottom-up, top-down, citizen... Maybe the author could clarify what made the cut giving more details, and what had low results.

Table 2 jumped to pg. 7.

Lines 200-208 font size changed.

Adjust table 5 title to same page.

Author Response

Thank you for your comments. I will work on the formatting issues that you identified. Regarding the topics that were identified, these are at a general (higher) level. So issues like mobility would fall under "transportation", participation is under "engagement", bottom-up and top-down are under "engagement" or "policy", etc. These distinct topics did not emerge despite using the multi-label approach. This is likely a linguistic issue with titles. A comment about this was added to the conclusions.

Reviewer 2 Report

General comments:

My main concern is the why? or so what? question, why classifying the topics and themes of research in urban planning is important for urban planning scholars? why knowing the most frequently cited topics is important? you need to address these questions clearly in the introduction and conclusion. in general, the research problem is not clear. the results of your study could be valuable to solve a research problem like what you mentioned in line #41 (the relationship between funding and scholarship activities)

Method:

The method you used is great, however, it lacks a lot of clarifications for the general reader, you need to include the mathematical equations, mention the software you used in your analysis, elaborate more on how did you elect these topics and labels. Most importantly, you need to elaborate on the limitations of the method in the method section.

Author Response

Thank you for your comments and suggestions. I added statements about the usefulness of the analysis to the introduction. In addition, I added to the methodology section which platform I used. Background on SVM is very involved so I encouraged readers to see detailed background on SVM in Joachims 1997 and Vapnick 2013.

Reviewer 3 Report

The author relates a topical and a citation analysis in the area of urban planning.

The paper is a good effort but needs some revisions before publication in "urban science" can be considered.

With respect to what determines citations of articles please also refer to Tahamtan and Bornmann (2019). Citation analysis with respect to topics is not so uncommon as is stated in the article. The keyword here is normalization of citations, which is also done within and not only across fields. See, e.g., Bornmann and Wohlrabe (2019). Please also refer to Gnewuch and Wohlrabe (2017), when it comes to title characteristics and citations. Please explain the basic idea how the topics in table 1 were identified. Just providing a link in a footnote is not enough. Please discuss the issue that the topics are in every case a real topic (at least in the referees opinion). "Analysis" is not a topic in a pure sense, almost all scientific articles are analysis in one way or the other. I understand that it comes out of the analysis but this should be discussed. How the SVM algorithm is explained is fine and sufficient but there is no information provided how the algorithm has been used here. How was the SMV trained? Or was it a unsupervised learning? Please provide more details. Please are also more precise what the advantage of the SVM over the a coword-occurance analysis is. Looking at Figure 1 one gets the impression that it was only checked if two topics ocurr (or mentioned) in the title. Please discuss or explain that there is no selection effect when only public GS profiles are used in the analysis. It could be the case that some important authors in the field do not have corresponding GS profile. Closely related to this issue: One could ask why does the author did not just downloaded urban planning articles from WOS and did the analysis? I am fine with GS, but you should discuss that issue. What about articles from the GS profiles that are not related to urban planning? I think many authors publish in different areas of science. So could it be the case that articles in the data set are not related to urban planning? Please provide some bounds (confidence bands) in Table 4 (as in Bornmann and Wohlrabe 2019). The cites per year (last column) are close. From my perspective it cannot be stated that the differ significantly (l. 263). The analysis has one major weakness, it does not control for other characteristics that influence the number of citations. The author does itself discuss this issue in the introduction but does nothing about it. It could be the case that the analysis might be biased. One suggestion: Please conduct a simple regression analysis which includes quantative title characteristics (length, number of authors, article age etc., see Gnewuch and Wohlrabe 2017) and dummies for the topics from table 4 (a dummy for every topic is not possible). You can than interpret the significance and size of the dummies after controlling for other factors. One important factor will be missing: the quality of the outlet (mostly journals) where the article has been published. An article in a prestigious journal gathers usually more citations than an article published in less well-known outlets. I suppose this would be difficult to realize in this context, but at least this issue should be discussed.

Literature

Lutz Bornmann & Klaus Wohlrabe, 2019. "Normalisation of citation impact in economics," Scientometrics, Springer;Akadémiai Kiadó, vol. 120(2), pages 841-884, August.

Matthias Gnewuch & Klaus Wohlrabe, 2017. "Title characteristics and citations in economics," Scientometrics, Springer;Akadémiai Kiadó, vol. 110(3), pages 1573-1578, March.

Iman Tahamtan & Lutz Bornmann, 2019. "What do citation counts measure? An updated review of studies on citations in scientific documents published between 2006 and 2018," Scientometrics, Springer;Akadémiai Kiadó, vol. 121(3), pages 1635-1684, December.

Author Response

Thank you very much for these detailed comments and suggestions. Based on these, the following points have been added to the article.

Reference to Tahantam and Bornmann 2019 added. Analyses of topics and citations are not uncommon to other disciplines, but have only been done in one other published case for planning (Stevens et al). Reference to Bornmann and Wohlrabe 2019 added. How topics were identified in Table 1 has been described in the methods section (Nvivo thematic analysis). Topics and article types somewhat mixed (e.g., issue with "analysis" as a topic). This is an artifact of the thematic analysis used for topic identification. Clarification on Figure 1 related to multiple labels has been added. Planning faculty with GS profiles are more highly cited compared to all planning faculty. This may be a source of bias, but the interest was in identifying the most highly cited, so this should not under-represent certain (highly cited) scholars. The focus is on what current urban planning faculty/scholars are publishing. Non-planners contribute to the planning literature, but the interest is in what those in planning faculty positions are publishing. In other words, what are planning programs (i.e., faculty) interested in. A limitation has been added about title and journal characteristics that have not been controlled for in determining citations per topic.

Round 2

Reviewer 2 Report

The article is improved tremendously

Author Response

Thank you again for your comments.

Reviewer 3 Report

Thanks for the revision.

Please cite Bornmann and Wohlrabe (2019) correctly as stated in my report. It is 2019 and not 2017, furthermore the issue and the journal pages must be corrected.

Author Response

Thank you again for your great comments.